# Feasibility of Brewer’s Spent Yeast Microcapsules as Targeted Oral Carriers

**DOI:** 10.3390/foods12020246

**Published:** 2023-01-05

**Authors:** Sofia F. Reis, Vitor J. Martins, Rita Bastos, Tânia Lima, Viviana G. Correia, Benedita A. Pinheiro, Lisete M. Silva, Angelina S. Palma, Paula Ferreira, Manuel Vilanova, Manuel A. Coimbra, Elisabete Coelho

**Affiliations:** 1REQUIMTE-LAQV, Department of Chemistry, University of Aveiro, 3810-193 Aveiro, Portugal; 2I3S, Institute for Research in Innovation in Health, University of Porto, 4200-135 Porto, Portugal; 3UCIBIO—Applied Molecular Biosciences Unit, Department of Chemistry, School of Science and Technology, NOVA University Lisbon, 2819-516 Caparica, Portugal; 4Associate Laboratory i4HB—Institute for Health and Bioeconomy, School of Science and Technology, NOVA University Lisbon, 2819-516 Caparica, Portugal; 5CICECO, Department of Materials and Ceramic Engineering, University of Aveiro, 3810-193 Aveiro, Portugal; 6IBMC, Institute for Molecular and Cell Biology, University of Porto, 4200-135 Porto, Portugal; 7ICBAS, School of Medicine and Biomedical Sciences, University of Porto, 4050-313 Porto, Portugal

**Keywords:** glucans, mannoproteins, in vitro digestion, innate immune system, Dectin-1, Dectin-2, DC-SIGN, carbohydrate microarrays, delivery systems

## Abstract

Brewer’s spent yeast (BSY) microcapsules have a complex network of cell-wall polysaccharides that are induced by brewing when compared to the baker’s yeast (*Saccharomyces cerevisiae*) microcapsules. These are rich in (β1→3)-glucans and covalently linked to (α1→4)- and (β1→4)-glucans in addition to residual mannoproteins. *S. cerevisiae* is often used as a drug delivery system due to its immunostimulatory potential conferred by the presence of (β1→3)-glucans. Similarly, BSY microcapsules could also be used in the encapsulation of compounds or drug delivery systems with the advantage of resisting digestion conferred by (β1→4)-glucans and promoting a broader immunomodulatory response. This work aims to study the feasibility of BSY microcapsules that are the result of alkali and subcritical water extraction processes, as oral carriers for food and biomedical applications by (1) evaluating the resistance of BSY microcapsules to in vitro digestion (IVD), (2) their recognition by the human Dectin-1 immune receptor after IVD, and (3) the recognition of IVD-solubilized material by different mammalian immune receptors. IVD digested 44–63% of the material, depending on the extraction process. The non-digested material, despite some visible agglutination and deformation of the microcapsules, preserved their spherical shape and was enriched in (β1→3)-glucans. These microcapsules were all recognized by the human Dectin-1 immune receptor. The digested material was differentially recognized by a variety of lectins of the immune system related to (β1→3)-glucans, glycogen, and mannans. These results show the potential of BSY microcapsules to be used as oral carriers for food and biomedical applications.

## 1. Introduction

Brewer’s spent yeast (BSY) is the second major by-product of the brewing industry [1] and a rich source of polysaccharides, mainly of mannoproteins and glucans, which may have a variety of food and biomedical applications [2,3,4,5].

The yeast (*Saccharomyces*) cell wall comprises up to 90% polysaccharides, accounting for 15–25% of the yeast cell dry weight [2]. The polysaccharides are mainly β-glucans linked through (β1→3)- and (β1→6)-d-Glc linkages, representing most yeast cell wall polysaccharides (55–65%), followed by mannoproteins (35–40%), glycogen (1–29%), and chitin, which account only for 1–2% of the yeast cell wall polysaccharides. Recently, the presence of (β1→4)-d-Glc linkages were also reported on the BSY cell wall structure; however, its importance to the rigidity and shape of the cell wall demonstrating linkage to the other polysaccharide structures is not clear [2,6,7,8,9]. Mannoproteins may be *N*-glycosylated, composed of 90% carbohydrates and 10% protein, with between 50 and 200 residues of (α1→6)-Man linked, highly branched by (α1→2)-Man and (α1→3)-Man terminals. The link to the protein is made through the amide group of asparagine of the peptide backbone and an *N*-acetylglucosamine (GlcNAc) residue linked to (α1→6)-Man. Mannoproteins may also be *O*-glycosylated, composed of 50% carbohydrates and 50% protein, with between one and five residues of (α1→2)-Man and (α1→3)-Man. The link to the protein is made through the hydroxyl group of serine or threonine of the peptide backbone. Glycogen is an α-glucan linked by (α1→4)-d-Glc and highly branched by (α1→6)-d-Glc linkages, whereas chitin is a linear polymer of (β1→4)-linked GlcNAc.

The cell wall polysaccharides have been solubilized from *Saccharomyces,* including BSY, by using different combinations of pretreatments (autolysis, sonication, high-pressure homogenization, hot water) followed by enzymatic hydrolysis [10,11,12] or alkaline extractions [7,9,13]. In all the cases, an insoluble material remains, which according to the solubilization process used, may account for up to 11–23% of the initial yeast or BSY. The carbohydrate composition may vary between 57 and 97%. This insoluble material, composed mainly of glucans that preserve the three-dimensional capsule-shaped structure of the initial yeast cell, is often known by yeast microcapsules or β-glucan particles [2]. In the case of BSY microcapsules, the three-dimensional structure is preserved due to the composition in (α1→4) and (β1→4) glucose linkages that, although resistant to an alkali extraction of up to 8 M [7], are partially degraded by cellulase and α-amylase [9]. They are also destroyed by hypochlorite due to the oxidation of the network promoted by the residual mannoproteins [8].

Yeast microcapsules have been used extensively as drug delivery systems [14,15,16,17,18,19] despite not being totally resistant to gastrointestinal tract (GIT) digestion [19]. This is due to the ability of β-glucans to enhance innate host defenses [20] and to the specific yeast microcapsules characteristics, including safety with no cytotoxic and genotoxic effects [10]. The binding efficiency of yeast microcapsules to macrophages via specific (β1→3)-glucan mammalian receptors can trigger immune cascade responses [20] and render yeast microcapsules a suitable target oral delivery system. The mechanisms involved in the immunological and inflammatory responses are complex and depend on the glucan’s characteristics (molecular weight, degree of branching, solubility, 3D conformation) and how they interact with their receptors [20]. Dectin-1, a major leukocyte receptor for (β1→3)-glucans, is a type II transmembrane protein expressed mainly at the cell surface of neutrophils, macrophages, and dendritic cells [20]. Dectin-1 also recognizes yeast microcapsules [16,17], enabling their internalization, which could be used for the targeted delivery of anti-inflammatory drugs.

Based on BSY microcapsules’ structural network, it is hypothesized that they may be used as oral delivery systems targeting the interaction of immune receptors due to their composition in (β1→3)-glucans, in line with yeast microcapsules. Additionally, its enrichment in α-glucans and the presence of residual mannoproteins may promote the interaction with other key immune receptors, which is an advantage when compared to yeast microcapsules. Thus, this work aims to study the resistance of BSY microcapsules to in vitro digestion (IVD), the influence on the recognition by the Dectin-1 immune receptor in HEK-Blue™ hDectin-1b reporter cell line after IVD, and the recognition of the IVD solubilized material by different mammalian immune receptors using a glycan microarray. The feasibility of BSY microcapsules as targeted oral carriers for food and biomedical applications is also discussed, considering these properties.

## 2. Materials and Methods

### 2.1. Brewer’s Spent Yeast Microcapsules Preparation

Brewer’s spent yeast (BSY) microcapsules resulted from four different extraction conditions: two alkaline and two subcritical water extractions applied to BSY gently supplied by Super Bock Group SA (Porto, Portugal). BSY (*Saccharomyces pastorianus*) was first submitted to an autolysis process which consisted of heating an aqueous suspension of BSY at 60 °C for 2 h. Then, the suspension was centrifuged (4696× *g*) at 4 °C for 20 min, and the residue was separated and washed with water three times. The washed BSY residue was then freeze-dried, stored in a vacuum desiccator, and later used.

#### 2.1.1. BSY Microcapsules Resultant from Alkaline Extractions

The freeze-dried washed BSY residue was boiled with 80% (*v*/*v*) ethanol for 10 min to obtain the alcohol-insoluble residue (AIR). The AIR was then extracted with 1 M KOH or 4 M KOH, each in 20 mM NaBH_4,_ using oxygen-free solutions to prevent peeling reactions. The alkali extractions were performed in a 1:60 (*w*/*v*) ratio at room temperature for 2 h with continuous stirring under an N_2_ atmosphere, and the insoluble material was at the end separated by centrifugation (24,700× *g*, 4 °C, 20 min). This insoluble material was then suspended in water, neutralized to pH 5–6 with glacial acetic acid, dialyzed, centrifuged (24,700× *g*, 4 °C, 20 min), and the final residue (BSY microcapsules) was freeze-dried and stored in a vacuum desiccator until further analysis.

#### 2.1.2. BSY Microcapsules Resultant from Subcritical Water Extractions (SWE)

The freeze-dried washed BSY residue was resuspended in distilled water in a ratio of 1:6 (*w*/*v*) and placed in a Teflon reactor for 4 min at 180 °C and in another batch for 2 min at 200 °C, under stirring in a microwave EthosSYNTH Labstation (maximum output, 1 kW, 2.45 GHz; Milestone Inc., Shelton, CT, USA). After extraction and depressurization, the suspensions were collected and centrifuged (24,700× *g*, at 4 °C, 20 min). The insoluble material (BSY microcapsules) was freeze-dried and stored in a vacuum desiccator until further analysis.

### 2.2. In Vitro Digestion (IVD)

The BSY microcapsules resultant from alkaline extractions (1 M KOH and 4 M KOH) and subcritical water extractions (180 SWE and 200 SWE) were submitted to static in vitro gastrointestinal digestion based on a reported method with some modifications [21]. The oral phase was simulated by using α-amylase type VI-B from porcine pancreas (Sigma Aldrich, St. Louis, MO, USA; A-3176-1MU) at pH 7 for 30 min. Briefly, α-amylase 21.5 U/mL of the sodium phosphate buffer 0.1 M, pH 6.9 was heated to 37 °C, and then the BSY microcapsules were added in a ratio of 1:20 (*w*/*v*) and left under stirring for 30 min at 37 °C. The gastric phase was simulated by using porcine pepsin (Sigma Aldrich; P-7000) at pH 1.5 for 1 h. First, the pH of the later solution was decreased by adding 2 M HCl, and then pepsin was added at 146 U/mL and left under stirring for 1 h at 37 °C. The intestinal phase was simulated by using porcine pancreatin (Sigma Aldrich; P-1750) at pH 7 for 3 h. The pH of the later solution was increased by adding 2 M NaOH, and then pancreatin was added at 200 mg/mL and left under stirring for 3 h at 37 °C. Finally, the solution was heated to 100 °C for 10 min to inactivate the enzymes before being centrifuged at 15,000 rpm for 20 min. The residue and supernatant were collected and dialyzed (12 kDa) in distilled water and then freeze-dried. Controls were performed with no addition of enzymes.

### 2.3. Chemical Composition Analysis of BSY Microcapsules before and after IVD

#### 2.3.1. Protein Content

Elemental analysis was used to determine the nitrogen content by using a Truspec 630-200-200 equipment (LECO Corporation, St. Joseph, MI, USA), combustion furnace temperature at 1075 °C, afterburner temperature at 850 °C, and the detection mode used was thermal conductivity. Protein content was estimated (N × 5.99) by taking into consideration the substantial composition of chitin in BSY [22].

#### 2.3.2. Sugar Analysis

Polysaccharide monomeric composition was determined by the neutral sugars released by acid hydrolysis and the analysis of their alditol acetates by gas chromatography [23] using a Perkin Elmer-Clarus 400 chromatograph (PerkinElmer, Waltham, MA, USA) with a split injector (split ratio 1:60) and an FID detector, according to the method described by the same authors [6].

#### 2.3.3. Glycosidic Linkage Analysis

Glycosidic linkage composition was determined by methylation analysis and analyzed by gas chromatography coupled to quadrupole mass spectrometry (GC-qMS) of the partially methylated alditol acetates (PMAA) [24] on a Shimadzu GCMS-QP2010 Ultra gas chromatograph (Shimadzu, Kyoto, Japan), equipped with a 25 m × 0.32 mm (i.d.), 0.1 µm film thickness HT-5 aluminum clad fused silica capillary column (SGE Analytical Science, Supelco, Bellefonte, PA, USA), according to the method described by the same authors [6].

### 2.4. Scanning Electron Microscopy (SEM) Analysis

The microstructures of the BSY insoluble material were analyzed by scanning electron microscopy (SEM) on a SU-70 Hitachi microscope (Hitachi, Tokyo, Japan) operated at 4 kV. Freeze-dried powder samples were deposited directly onto a double-sided carbon conductive tape glued to an aluminum holder. The excess material was removed, and a conductive carbon thin film was deposited onto the powders using a carbon rod coater (Emitech K950X; Quorum Technologies, Lewes, UK).

### 2.5. Stimulation of Human Dectin-1 Receptor

HEK-Blue™ hDectin-1b reporter cell line (InvivoGen, San Diego, CA, USA) was used to study the stimulation of human Dectin-1 receptors by BSY microcapsules after IVD. These reporter cells were engineered to express high levels of human Dectin-1b isoform and genes involved in the Dectin-1/NF-κB/SEAP signaling pathway. Activation was then determined according to the levels of secreted embryonic alkaline phosphatase (SEAP), which was measurable through a colorimetric assay upon reaction with QUANTI-Blue™. HEK Blue™ hDectin 1b cells do not respond to soluble β-glucans.

HEK-Blue™ hDectin-1b cells were grown in a DMEM culture medium supplemented with 4.5 g/L glucose (Sigma-Aldrich), 10% of heat-inactivated Fetal Bovine Serum Premium (FBS) (Biowest, Nuaillé, France), 2 mM l-glutamine, 100 U/mL penicillin, 100 μg/mL streptomycin (all from Sigma-Aldrich), and HEK-Blue™ selection antibiotics: 100 μg/mL Normocin™ and 1 µg/mL puromycin (InvivoGen). Cells were grown in vented T-75 flasks and incubated at 37 °C in 5% CO_2_. The growth medium was renewed 2 times a week until the culture reached 70–80% confluency. HEK-Blue™ hDectin-1b cells were rinsed and detached from the T-75 flask, re-suspended in the HEK-Blue™ Detection medium, and seeded into flat-bottom 96 well-culture plates at a concentration of 5 × 10^4^ cells/well. Cells were stimulated with 20 µg/mL of BSY microcapsules after IVD, 20 µg/mL of WGP^®^-Soluble (yeast soluble β1,3/β1,6-d-glucan), and 20 µg/mL of Zymosan (*Saccharomyces cerevisiae* particulate (β1→3)-d-glucan with 240 kDa) as a positive control. Sterile PBS was used as a negative control. The plates were incubated at 37 °C in 5% CO_2_ for 16 h. Upon reaction with QUANTI-Blue™, SEAP levels were measured by reading the optical density at 655 nm in a microplate reader Biotek™ Gen5™ (Thermo Fisher Scientific, Waltham, MA, USA).

The experiments were performed at least in triplicate (*n* ≥ 3). Data were reported as means ± SD and analyzed by one-way ANOVA and Tukey’s post hoc test using GraphPad Prism (Version 9.41; GraphPad Software, San Diego, CA, USA). Results were considered statistically significant if *p* < 0.05.

### 2.6. Carbohydrate Microarray Construction and Analysis

The information given on the polysaccharide samples, microarray construction, imaging, and data analysis are according to the Minimum Information Required for A Glycomics Experiment (MIRAGE) guidelines for reporting glycan microarray-based data [25].

#### 2.6.1. Polysaccharides and Microarray Construction

The soluble material obtained after the in vitro digestion of BSY microcapsules was analyzed for their recognition by different proteins in a carbohydrate microarray format. For comparison purposes, eight glucans and mannans were included as controls to increase the carbohydrate structural diversity of the array. The list of polysaccharides, their sources, and their main composition are given in Appendix A. For the construction of the microarray, the polysaccharides were dissolved in Milli Q water with 0.02% (*w*/*v*) NaN_3_ at a final concentration of 0.5 mg/mL and immobilized by non-covalent means onto 2-pad nitrocellulose coated glass slides (UniStart^®^, Sartorius Stedim Biotech, Göttingen, Germany) using a MicroCasterTM 8-Pin System (Whatman^®^, Maidstone, UK). The aqueous solutions contained Cy3 NHS ester (GE Healthcare, Chicago, IL, USA) at 20 ng/mL as a marker to monitor the printing process. Each polysaccharide was printed in triplicate (20–70 nL per spot with approximately 500 μm diameter each) at an ambient temperature.

#### 2.6.2. Microarray Binding Analysis

The microarray was probed for recognition with five carbohydrate-binding proteins with reported specificities, including 3 mammalian immune receptors (human Dectin-1 and DC-SIGN and mouse Dectin-2), 1 plant lectin (Concanavalin A, ConA), and 1 monoclonal antibody (400-2 anti-β1,3-glucan). Detailed information on their names, sources, conditions of analysis, and reported carbohydrate specificity is in Appendix A. The microarray binding analyses were performed essentially as described [26] and specifically according to the method described by the same authors [6]. The analysis performance and the slide’s drying and storage were at ambient temperature. The quantitation of microarray data was carried out using the GenePix^®^ Pro Software 7.3.1 (Molecular Devices, San Jose, CA, USA).

## 3. Results and Discussion

To evaluate the feasibility of oral carriers of microcapsules obtained after alkali and subcritical water extractions, their resistance to in vitro digestion (IVD) was assessed. In addition, recognition experiments were performed for the non-digested microcapsules with the Dectin-1 immune receptor and the solubilized polysaccharides with other carbohydrate-binding proteins from the innate immune system.

### 3.1. Resistance of Microcapsules to In Vitro Digestion (IVD)

The in vitro digested material accounted for 44–63% of the initial BSY microcapsules, with the highest values observed for the ones obtained from alkaline extractions, in particular the 1 M KOH (Figure 1). The microcapsules obtained after subcritical water extraction (SWE) were shown to be more susceptible to IVD, probably due to the strong extraction conditions used in the microcapsule’s preparation. However, only with the digestion medium solutions, without the enzymes (controls), the BSY microcapsules obtained from SWE showed the release of between 33% and 27% for 180 SWE and 200 SWE, respectively, corresponding to only 21% and 17% of the enzymes digested material. This is less than half of the values obtained for 1 M KOH (41%) and 4 M KOH (44%), which means that carbohydrates and proteins are likely to be more accessible in the microcapsule structures obtained from alkaline extractions than from SWE. The subcritical water treatments probably promote Maillard reactions that modify polysaccharides [27,28] and prevent their enzymatic degradation.

Almost half of the initial BSY microcapsules remained non-digested, independently of the conditions used. Their spherical shape morphology before and after IVD was preserved, despite some visible agglutination and deformation (Figure 2). 

### 3.2. Influence of In Vitro Digestion (IVD) on the Cell-Wall Polysaccharides Structure

The protein and carbohydrate composition of BSY microcapsules before and after IVD revealed that in the case of alkaline extractions, half of the protein was digested by the digestive enzymes, enriching the microcapsules in carbohydrates (Table 1). The molar % of Man and Glc was not affected, which suggests that mannoproteins from the cell wall were not being released. However, the glycosidic linkages increment on t-Man, 2-Man, and 2,3-Man after IVD on both BSY microcapsules resultant from the alkaline extractions (Table 2) suggests that these microcapsules were enriched in linear *O*-glycosylated due to the solubilization of the branched *N*-glycosylated mannoproteins. Moreover, the decrease in 4-Glc and 4,6-Glc and the increase in 3-Glc after IVD (Table 2) indicates that the digestive enzymes also solubilized glycogen, enriching the BSY microcapsules in (β1→3)-glucans together with *O*-glycosylated mannoproteins.

In the case of BSY microcapsules from SWE resultant from the 180 °C MAE treatment, there was a slight decrease in the protein content after IVD (from 32 to 26%) and a decrease in the carbohydrates by the effect of the *digesta* medium (from 41 to 28%). The molar % of Man and Glc decreased from 23 to 5 mol% of the *digesta* medium (Table 1) meant that it could be inferred that mannoproteins were digested only with acidic conditions. This was also confirmed by the decrease in 2-Man and 3-Man (Table 3) after IVD and in the controls, which are the branched side chains of (α1→6)-linked mannan [2] and are more susceptible to acidic degradation. In addition, 4-Glc linkages decreased after IVD and in the controls, also enriching the microcapsules in (β1→3) glucans. However, it seems that these linkages are not from native glycogen when looking at the trace values of 4,6-Glc (Table 3) and are more likely to be debranched glycogen or the recently reported (β1→4)-d-Glc linkages which are not yet deeply explored [2].

The BSY microcapsules from SWE, resultant from the 200 °C MAE treatment, were enriched in protein and carbohydrates after IVD, which means that most of the material released by digestion was not quantified as protein and carbohydrates but were probably their modified structures, such as melanoidins, due to the overstated heat treatment applied to the BSY [27]. Indeed, most of this material was solubilized only by the *digesta* medium solutions, as seen by the increase in the content of protein and carbohydrates in the controls (Table 1). While 44% of the material solubilized was resultant of the action of IVD, only 7% was resultant of the action of the digestive enzymes (Figure 1). The proportion between the content of mannoproteins and glucans was similar, allowing the inference that it was not affected by the IVD or by the *digesta* medium solutions (Table 1). Similarly to the microcapsules obtained at 180 °C (180 SWE), these BSY microcapsules obtained with subcritical water at 200 °C had a lower content in mannoproteins, confirmed by the decrease in 2-Man linkages and the enrichment in (β1→3) glucans due to the decrease in 4-Glc linkages (Table 3).

The microcapsules obtained after 200 °C SWE seem to contain (1→4)-linked glucans resistant to digestion. The low amount of 4,6-Glc (0.7 mol%) may allow the inference that (1→4)-linked glucans were mostly linear and not accessible to α-amylase due to the presence of (β1→4)-linked glucans and Maillard reaction products formed under high-temperature conditions [27]. However, transglycosylation reactions cannot be excluded [27,28]. The presence of modified material to a greater extent than the 180 °C SWE microcapsules can also be deduced by the presence of the residue 2,3,4,6-Glc, which before IVD represented 5.9 mol% of the glycosidic-linkages composition and after IVD represented 34.8 mol%. After the enzymatic digestion of the proteins and glycogen, the structures that remained in the microcapsules were those modified and consequently not hydrolyzed. These structures may be resistant glycogen in line with the formation of resistant starch due to heat treatment [29].

### 3.3. Recognition of Microcapsules by Dectin-1 Immune Receptor

The material not solubilized by IVD from the BSY microcapsules was analyzed for recognition by the human Dectin-1 immune receptor measured by Dectin-1/NF-κB activation in the HEK-Blue hDectin-1b cell line (Figure 3). All the samples showed a cell recognition higher than the negative control (soluble β-glucans). The BSY microcapsule resultant from 1 M KOH after IVD were the only ones with activity like the positive control (zymosan). BSY microcapsules resulting from 4 M KOH after IVD were recognized similarly to those that resulted from 1 M KOH, despite showing lower Dectin-1 recognition than zymosan. Both BSY microcapsule samples, resulting from SWE, presented a similar Dectin-1 recognition that was lower than the ones obtained from alkaline extractions and zymosan.

The samples were all rich in glucans, particularly (β1→3) glucans, yet the sample from 1 M KOH was not the one with the highest content in 3-Glc linkages (Table 2) but was the one with the highest branched structures, as seen by the proportion of 3-Glc:6-Glc:3,6-Glc (19:2:1). Indeed, Dectin-1 is a non-opsonic phagocytic pattern-recognition receptor that recognizes not only linear (β1→3) but also those branched with (β1→6)- linked glucans [30]. The samples resultant from 180 SWE presented the highest value of 3-Glc linkages (Table 3) but also presented the lowest branched structures, 3-Glc (39):6-Glc (1):3,6-Glc (1), while samples from 200 SWE presented the lowest value of 3-Glc linkages and the higher branched structures, 3-Glc (27):6-Glc (1):3,6-Glc (1), which may explain their similar Dectin-1 recognition.

### 3.4. Characterization of Digested Material from Microcapsules after In Vitro Digestion (IVD)

The digested material (>12 kDa) from BSY microcapsules after IVD is mostly composed of proteins and carbohydrates (Table 4), including the material solubilized only by the *digesta* medium solution except in the case of 4 M KOH; yet the yield of solubilization is too low (9%), as shown in Figure 1.

The material digested from the BSY microcapsules and resultant from alkaline extractions was consistent with glycogen, as seen by the major linkages present in 4-Glc and 4,6-Glc and, also *O*-glycosylated mannoproteins seen by the values of t-Man, 2-Man, 3-Man and the absence of 6-Man linkages (Table 5) confirming the suggestion on Section 3.2. Additionally, looking at the balance of Man and Glc (Table 4), the t-Man, 2-Man, 4-Glc, and t-Glc values (Table 5) and the yields (Figure 1) between the digested material > 12 kDa and the controls, it seems that the low pH of the *digesta* solutions solubilizes *O*-glycosylated mannoproteins and glycogen and the action of the digestive enzymes turned them into small polymers. These small polymers, in the case of glycogen, are lost in the dialysis process, meaning that the branched structures of glycogen remain in the high molecular weight material (>12 kDa).

The presence of glycogen was expected since insoluble glycogen covalently linked to the cell wall (β1→3)-d-glucans through a (β1→6)-linkage was already reported [2]. However, the presence of mannoproteins is quite interesting. As they were usually reported to be placed at the outer layer of the cell wall, their removal from the BSY during the extraction process could be expected. However, it seems that these structures are also located deep in the structure of the cell wall and are not removed by alkaline extraction [2]. This fact strengthens the hypothesis that the three-dimensional shape of the yeast cell wall is preserved by a network of glucans and proteins and that *O*-glycosylated mannoproteins have an important role in this. The presence of glycogen after IVD may contribute to glucose uptake, and the resistant glycogen may also be seen as an add since it may act as dietary fiber and/or a prebiotic.

In the case of the material digested from the BSY microcapsules and resultant from SWE using MAE treatment, the digested material >12 kDa beyond *O*-glycosylated mannoproteins may also present a residual amount of *N*-glycosylated mannoproteins, as seen by the residual amount of 6-Man (Table 6). However, the value is low (0.5 mol%) and may be a consequence of heat treatment since this position is the susceptible to transglycosylation reactions [27,28]. Regarding glycogen, it seems to be present in residual amounts as seen by the molar % of 4,6-Glc (Table 6) but, in contrast with the digested material from BSY microcapsules resultant from alkaline extractions, a considerable amount of 3-Glc is present, reflecting the solubilization of (β1→3) glucans and are maybe linked to some of the 4-linked glucans (Table 6).

These results reflect that SWE using MAE is more efficient in extracting glycogen and mannoproteins from the BSY cell wall than alkaline extractions and that *O*-glycosylated mannoproteins have an important role in the three-dimensional shape of the yeast cell wall, as seen by the digested material after IVD, which presents some deformation of the microcapsules and others still retain their spherical shape (Figure 2b).

### 3.5. Recognition of Digested Material by Carbohydrate-Binding Proteins

A focused carbohydrate microarray was constructed containing the solubilized material (>12 kDa) after IVD from the BSY microcapsules resulting from alkaline extractions and SWE. Controls were also included, such as glucans and mannans isolated from other sources (Appendix A). The microarray was probed for recognition with the following mammalian immune receptors: human Dectin-1 (hDectin-1-Fc), human DC-SIGN (hDC-SIGN-Fc), and mouse Dectin-2 (mDectin-2-His). Other carbohydrate-binding proteins with known carbohydrate-binding specificity were also analyzed and included the anti-(β1→3)-glucan monoclonal antibody (mAb) 400-2, and α-mannose specific plant lectin Concanavalin A (ConA) (Appendix A).

Microarray analysis showed that only the polymeric material released from BSY microcapsules resulting from SWE was recognized by human Dectin-1 (Figure 4). These corresponded to samples with a considerably higher molar % of 3-Glc linkages (Table 6) when compared with the polymeric material released from BSY microcapsules and the resultant from alkaline extractions (Table 5). This immune receptor is known to be highly specific for (β1→3)-gluco-oligosaccharide sequences, with a minimum chain length of 10 monosaccharides, and is likely to recognize a helical conformational epitope on the digested material [31] (Appendix A). The differential presence of (β1→3)-glucans in the different samples was corroborated by the analysis with (β1→3)-glucan specific mAb 400-2.

Dectin-2 is known to recognize the core (α1→2)-linked *N*-mannan triantennary structure of the fungal cell wall [32] (Appendix A), and accordingly, the binding signals elicited by the samples (Figure 4) corroborate the molar % of mannoproteins, assessed by the structural analyses (Table 5 and Table 6).

The immune receptor DC-SIGN exhibits a broader binding profile to mannans and glucans [32] (Appendix A); however, the binding signals elicited by the samples analyzed (Figure 4) seem to be directly correlated to the molar % of glycogen (Table 5 and Table 6).

The plant lectin ConA is known to have specificity towards α-mannosylated glycans (Appendix A), and the elicited binding signals to the samples do not reflect the mannoproteins content assessed by the structural analyses (Table 5 and Table 6); other parameters, such as molecular weight, degree of branching, solubility, and 3D conformation, not measured in this work, are affecting the result.

The microarray results showed that the polymeric material released from BSY microcapsules, resulting from alkaline extractions and SWE in different conditions, was differentially recognized by a variety of lectin receptors of the immune system.

### 3.6. Feasibility of BSY Microcapsules as Targeted Oral Carriers

BSY microcapsules were already isolated as a source of β-glucans [11], but they were never used as oral delivery systems to the best of our knowledge. The results of this work have shown that BSY microcapsules are a potential targeted oral carrier. Like yeast microcapsules, they are resistant to IVD digestion and recognized by the Dectin-1 immune receptor due to the composition of (β1→3)-glucans and are thus able to act as a targeted oral carrier. In addition, the material solubilized during IVD may interact with different immune receptors, depending on the methodology used for microcapsule preparation, improving innate immunity due to their immunomodulatory properties. The soluble polysaccharides released during IVD from microcapsules obtained by SWE interact with Dectin-1, those released from the 1 M KOH microcapsules interact with DC-SIGN, and 4 M KOH and 180 SWE microcapsules solubilized polysaccharides interact with Dectin-2.

The following studies should proceed with a load of bioactive compounds for food and biomedical applications. From the perspective of biomedical applications and in line with the studies reported for yeast microcapsules, but at reduced costs, they may be used for the delivery of anti-inflammatory drugs [16,17,18,33] or specific therapies to different inflammation sites [19], where immune cells are recruited by the immune system (usually the sites of disease needed to be treated). An increase in the treatment efficacy is also expected to occur due to the increase in digestion resistance and the possibility of tuning different immune responses. From the perspective of food applications, being safe and resistant to digestion are qualities that mean they may be used to produce new food products with added health benefits, with the incorporation of bioactive molecules sensible to the digestion conditions. The addition of loaded BSY microcapsules to foods will claim both the benefit provided by the encapsulated bioactive and the immunomodulatory properties provided by the solubilized material during digestion.

## 4. Conclusions

The BSY insoluble material resultant from different processing conditions and composed of microcapsules of a spherical shape were submitted to in vitro digestion (IVD), and half of the material was digested; nevertheless, the non-digested material, despite some visible agglutination and deformation of the microcapsules kept their spherical shape. The digested material varied according to the composition of the initial BSY microcapsules, but in all cases, the non-digested material was enriched in (β1→3)-glucans.

It was observed that the BSY microcapsules resultant from alkaline extractions were composed of glycogen and *O*-glycosylated mannoproteins which were readily digested, while in the BSY microcapsules, resultant from SWE, glycogen was residual. Thus, in addition to the *O*-glycosylated mannoproteins, (β1→3)-glucans and (1→4)-glucans (excluding native glycogen) were also solubilized in the BSY microcapsules resultant from SWE.

The enriched (β1→3)-glucan BSY microcapsules were all recognized by human Dectin-1, and the digested material was recognized by a variety of lectin immune receptors. These results indicate the potential of BSY microcapsules to be used as carriers for oral delivery systems since they can simultaneously deliver bioactive compounds while targeting the interaction of immune receptors and stimulating the innate immune system of the host.

## Figures and Tables

**Figure 1 foods-12-00246-f001:**
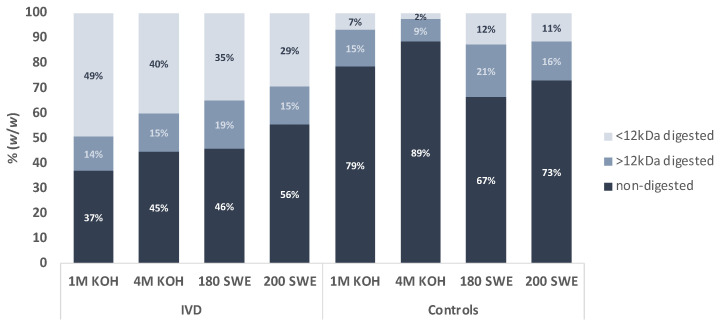
Percentage (%) of digested (<12 kDa and >12 kDa) and non-digested material of alkaline and subcritical water BSY microcapsules after in vitro digestion (IVD) and respective controls.

**Figure 2 foods-12-00246-f002:**
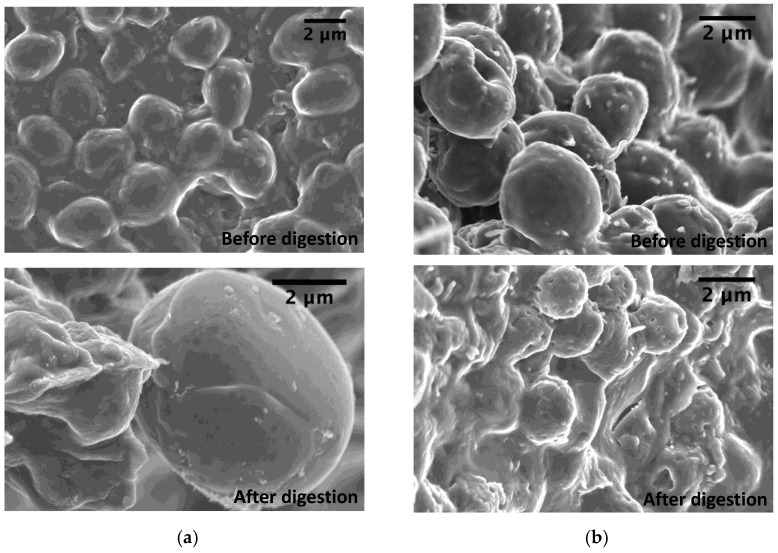
Scanning electron microscopy (SEM) images of alkaline (**a**) and subcritical water BSY microcapsules (**b**) before and after in vitro digestion.

**Figure 3 foods-12-00246-f003:**
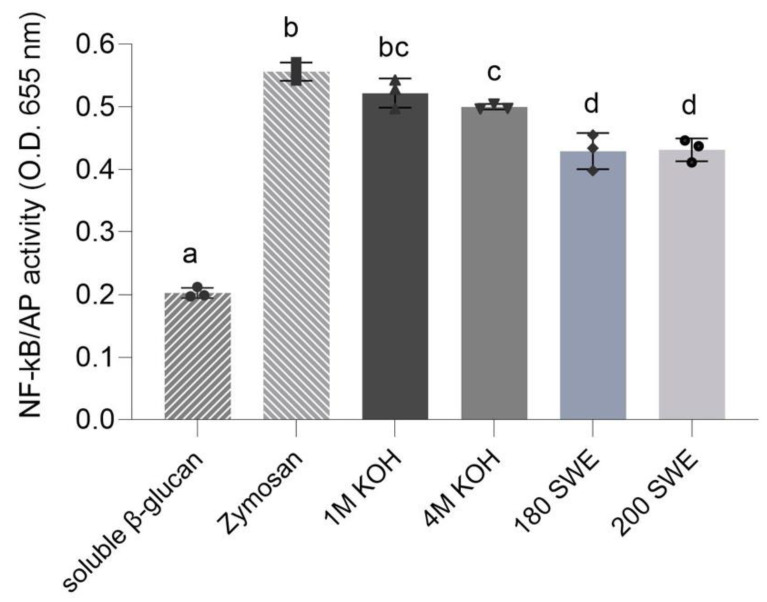
Recognition of alkaline and subcritical water BSY microcapsules after in vitro digestion by the human Dectin-1b receptor expressed by the activity of NF-κB in the HEK-Blue hDectin-1b cell line. Soluble β-glucan and zymosan used as negative and positive controls, respectively. Bars correspond to means ± SD. Means with different superscript letters are significantly different (*p* < 0.05), as analyzed by one-way ANOVA and Tukey’s test.

**Figure 4 foods-12-00246-f004:**
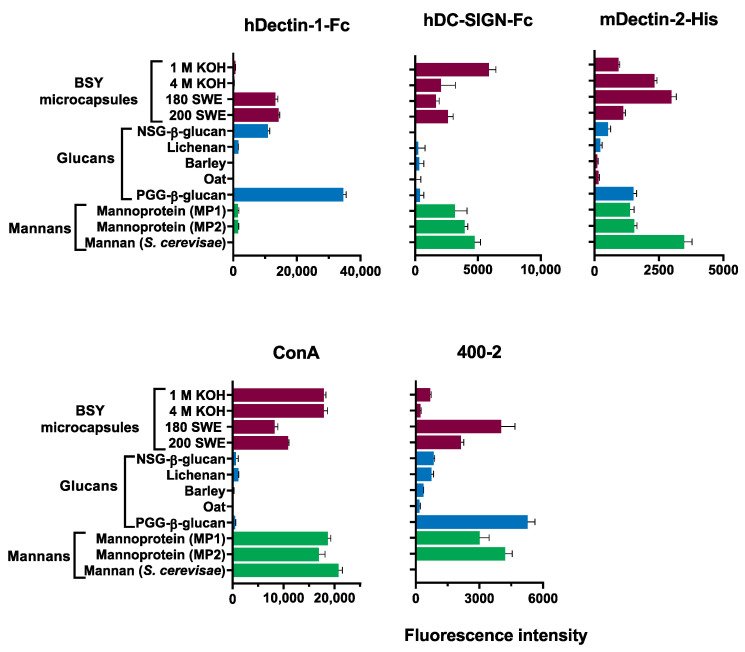
Carbohydrate microarray analyses of the binding to digested material (>12 kDa) from alkaline and subcritical water BSY microcapsules by carbohydrate sequence-specific proteins. The binding to digested alkaline and subcritical water BSYIM are represented by wine bars. Glucans (blue bars) and mannans (green bars), isolated from other sources, were included as controls for the binding. Detailed information on the polysaccharide samples and proteins analyzed is in Appendix A, respectively. The scales for the fluorescence binding intensities are depicted at the bottom for each protein; these are means of fluorescence intensities of triplicate spots arrayed (with error bars) at 0.5 mg/mL per spot. Mammalian immune receptors: hDectin-1-Fc—Human Dectin-1; hDC-SIGN-Fc—Human DC-SIGN; mDectin-2-His—Mouse Dectin-2. Plant lectin: ConA—Concanavalin A. Monoclonal antibody: 400-2 (anti-(β1,3)-glucan).

**Table 1 foods-12-00246-t001:** Carbohydrate and protein content of alkaline and subcritical water BSY microcapsules before and after in vitro digestion and respective controls.

BSY Microcapsules		Protein (Total %)	Carbohydrates (Total %)	Man mol %	Glc mol %
1 M KOH	Before IVD	31	58	9	91
	After IVD	15	79	6	94
	Control	24	57	7	93
4 M KOH	Before IVD	32	61	14	86
	After IVD	13	79	13	87
	Control	30	46	9	91
180 SWE	Before IVD	32	41	23	77
	After IVD	26	44	4	96
	Control	46	28	5	95
200 SWE	Before IVD	32	13	40	60
	After IVD	56	42	30	70
	Control	60	41	46	54

IVD—in vitro digestion; SWE—subcritical water extraction.

**Table 2 foods-12-00246-t002:** Glycosidic linkage composition of alkaline BSY microcapsules before and after in vitro digestion and controls.

Linkage	1 M KOH			4 M KOH		
	Before IVD	After IVD	Control	Before IVD	After IVD	Control
t-Man	3.7	6.2	3.0	8.9	11.2	7.0
2-Man	0.7	1.9	1.1	0.7	3.6	2.7
3-Man	0.3	------	0.2	0.3	0.8	0.8
6-Man	------	------	0.1	0.2	------	0.4
2,3-Man	0.1	3.0	0.3	0.7	1.6	0.9
2,6-Man	------	0.5	1.2	2.0	0.6	1.3
2,3,4,6-Man	0.2	0.6	0.1	0.7	0.7	0.2
Total Man	5.1	12.2	6.0	18.0	18.5	13.2
t-Glc	11.7	14.2	11.1	9.2	12.4	10.5
3-Glc	18.5	57.1	15.0	20.0	55.6	22.6
4-Glc	57.6	1.6	60.3	40.7	2.1	45.0
6-Glc	1.4	5.8	2.8	3.8	4.9	3.5
3,6-Glc	1.3	3.3	1.1	1.3	2.8	1.8
4,6-Glc	3.3	0.1	2.6	3.8	0.1	2.0
2,3,4,6-Glc	1.0	4.8	0.9	2.5	2.9	1.2
Total Glc	94.8	86.8	93.7	81.5	80.7	86.7

IVD—in vitro digestion.

**Table 3 foods-12-00246-t003:** Glycosidic linkage composition of subcritical water BSY microcapsules before and after in vitro digestion and controls.

Linkage	180 SWE			200 SWE		
	Before IVD	After IVD	Control	Before IVD	After IVD	Control
t-Man	9.9	1.7	2.9	15.3	5.0	6.9
2-Man	6.4	0.6	------	7.7	1.7	------
3-Man	1.6	------	------	1.3	------	------
6-Man	0.8	------	------	1.5	------	------
2,3-Man	0.2	0.3	0.9	------	------	------
2,6-Man	6.3	0.1	0.4	9.6	0.7	0.6
2,3,4,6-Man	0.2	0.3	0.7	1.4	4.5	2.3
Total Man	25.5	2.9	5.0	36.9	11.9	9.8
t-Glc	8.7	10.5	13.8	8.7	4.7	5.4
3-Glc	41.0	77.4	61.6	17.7	27.0	22.5
4-Glc	13.9	1.5	4.7	21.6	12.5	24.8
6-Glc	5.8	1.6	5.6	4.0	1.2	------
3,6-Glc	3.6	2.0	3.6	1.2	1.4	1.8
4,6-Glc	0.2	tr	0.3	1.6	0.7	1.4
2,3,4,6-Glc	0.7	3.4	4.3	5.9	34.8	29.5
Total Glc	73.9	96.4	93.9	60.6	82.3	85.4

IVD—in vitro digestion; SWE—subcritical water extraction.

**Table 4 foods-12-00246-t004:** Carbohydrate and protein content of polymeric material released from alkaline and subcritical water BSY microcapsules after in vitro digestion and respective controls.

BSY Microcapsules		Protein (Total %)	Carbohydrates (Total %)	Man mol %	Glc mol %
1 M KOH	Digested > 12 kDa	39	35	42	58
	Control	36	52	30	70
4 M KOH	Digested > 12 kDa	43	44	67	33
	Control	20	32	36	64
180 SWE	Digested > 12 kDa	43	33	68	32
	Control	33	49	56	44
200 SWE	Digested > 12 kDa	52	23	54	46
	Control	49	28	46	54

IVD—in vitro digestion; SWE—subcritical water extraction.

**Table 5 foods-12-00246-t005:** Glycosidic linkage composition of polymeric material released from alkaline BSY microcapsules after in vitro digestion and respective controls.

Linkage	1 M KOH		4 M KOH	
	Digested > 12 kDa	Control	Digested > 12 kDa	Control
t- Man	22.7	13.4	34.3	15.4
2-Man	4.8	4.0	7.3	4.0
3-Man	1.5	1.4	2.9	1.4
6-Man	------	------	------	------
2,3-Man	------	------	------	------
2,6-Man	1.6	2.1	0.8	1.5
2,3,4,6-Man	0.7	0.1	1.0	0.2
Total Man	31.3	21.0	46.2	22.4
t-Glc	25.9	8.5	17.8	8.3
3-Glc	0.2	0.2	0.3	0.9
4-Glc	29.1	62.6	22.4	60.4
6-Glc	2.3	2.3	2.6	4.0
3,6-Glc	1.3	0.5	1.1	0.5
4,6-Glc	7.1	3.9	6.0	2.2
2,3,4,6-Glc	2.6	1.1	3.2	1.4
Total Glc	68.5	79.0	53.4	77.6

**Table 6 foods-12-00246-t006:** Glycosidic linkage composition of polymeric material released from subcritical water BSY microcapsules after in vitro digestion and respective controls.

Linkage	180 SWE		200 SWE	
	Digested > 12 kDa	Control	Digested > 12 kDa	Control
t-Man	44.4	29.3	32.1	54.7
2-Man	10.9	7.0	10.4	4.7
3-Man	3.4	1.7	1.9	2.2
6-Man	0.5	------	0.4	------
2,3-Man	------	------	------	------
2,6-Man	1.1	1.4	1.4	1.8
2,3,4,6-Man	1.2	1.5	1.0	1.8
Total Man	61.5	40.9	47.1	65.1
t-Glc	6.1	10.6	8.0	16
3-Glc	15.9	13.2	26.7	14
4-Glc	5.3	22.1	7.0	4.9
6-Glc	2.0	5.9	2.0	------
3,6-Glc	0.9	0.9	1.3	------
4,6-Glc	1.0	2.7	1.1	------
2,3,4,6-Glc	6.9	4.5	6.1	1.8
Total Glc	38.1	59.8	52.3	36.6

SWE—subcritical water extraction.

## Data Availability

All relevant data are within the manuscript and its Supporting Information files.

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
