# Peer review of "Feasibility of Brewer’s Spent Yeast Microcapsules as Targeted Oral Carriers"

_foods, 2023, doi:10.3390/foods12020246_

Round 1

Reviewer 1 Report

In my opinion, the study described in the present manuscript is well though out and thorough, albeit with somewhat overstated links between the valid conclusions from the work and further applications in food. Some specific comments I would like the authors to address:

1) The BSY structures are referred to as microcapsules, yet they simply represent insoluble fractions obtained under different conditions. The microcapsule implies that you have a clear wall material/sheath that encloses an interior compartment. If this is not satisfied, the BSY structures would be better defined as microparticles. Can the authors comment on this?

2) The link between the results described and the suitability of the BSY structures for application as oral carriers in food is somewhat tenuous in the manuscript. In the final section of the introduction, the authors write "The feasibility of BSY microcapsules as oral carriers for food and biomedical applications is also discussed considering these properties." However, this is not discussed in any detail later in the manuscript. What makes these structures suitable for use in food, what type of bioactive would they be suited for, how would the BSY structures help in either protecting, compatibilizing or providing controlled release in a food matrix, etc.

Reviewer 2 Report

1.   On the basis of the data presented in the article, there can be no absolute confidence that the term microcapsule is appropriate for the material under investigation. Without defining the characteristics of the core or the internal reservoir, it is appropriate to define the material under investigation in the same way as analogous materials are defined in the cited articles, as glucan particles or nanoparticles (or microparticles depending on size) [14-17].

2.   The introduction does not sufficiently disclose the targeted microparticle delivery materials which are planned without polysaccharides (unless they are the only one bioactive material to be delivered), as well as the biomedical target and its intended location: inside the gut, in the intestinal layers, or behind the intestines. The qualitative characteristics that would be required for the assessment of transfer depend on this definition. In the absence of a clear definition and the selection of appropriate targeting methods to control the transfer to the target site, it is not appropriate to consider particles as target carriers of an active ingredient.

3.   If the purpose of the application is exclusively nutritional then the claims in the test and the conclusion should be revised and clarified together with a critical evaluation of the immune stimulatory effects.

4.   In view of the objectives, the methodological design and results of the paper are suitable, following corrections or additions in the light of comments 1-3.
